# Few-shot Cross-domain Image Generation via Inference-time Latent-code Learning

**Arnab Kumar Mondal***, **Piyush Tiwary**[†], **Parag Singla*** **& Prathosh AP**[†]
* IIT Delhi, [†] IISc Benglauru

## Abstract

In this work, our objective is to adapt a Deep generative model trained on a large-scale source dataset to multiple target domains with scarce data. Specifically, we focus on adapting a pre-trained Generative Adversarial Network (GAN) to a target domain without re-training the generator. Our method draws the motivation from the fact that out-of-distribution samples can be 'embedded' onto the latent space of a pre-trained source-GAN. We propose to train a small latent-generation network during the inference stage, each time a batch of target samples is to be generated. These target latent codes are fed to the source-generator to obtain novel target samples. Despite using the same small set of target samples and the source generator, multiple independent training episodes of the latent-generation network results in the diversity of the generated target samples. Our method, albeit simple, can be used to generate data from multiple target distributions using a generator trained on a single source distribution. We demonstrate the efficacy of our surprisingly simple method in generating multiple target datasets with only a single source generator and a few target samples. The code of the proposed method is available at: `https://github.com/arnabkmondal/GenDA`

## 1 Introduction

### 1.1 Few Shot Image Generation

Deep generative models learn to generate novel data points from an unknown underlying distribution. The family of auto-encoder based generative models (Kingma & Welling, 2014) use variational inference to maximize evidence lower bound (ELBO) on the data likelihood; adversarial generators such as GANs (Goodfellow et al., 2014) learn to sample by solving a min-max optimization game and the normalizing flow-based methods utilize tractable transformations between the latent and data distributions (Kobyzev et al., 2020). All such models are shown to be successful in generating high-quality realistic data such as images (Karras et al., 2018; 2019; 2020b).

However, one of the caveats in deep generative models is that they require thousands of images for proper training, limiting the scope of what can be explored (Sushko et al., 2021). This problem poses practical restrictions on the applications of deep generative models, as the number of training data is often limited to the order of hundreds or even tens at times; making it crucial to adapt generative models for few-shot settings. One natural way to accomplish the above objective is to use the 'prior-knowledge' that is already there in a generative model built on a larger, but 'close' source dataset (Wang et al., 2018; 2020b). Several ideas ranging from learning latent transformations (Wang et al., 2020b) to re-training generators on target data with regularizers such as Elastic-weight consolidation (Li et al., 2020) and cross-domain correspondence (Ojha et al., 2021) have been proposed (See section 2 for a detailed description). The basic principle in all these is to adapt the generator of Generative Adversarial Network (GAN) (Goodfellow et al., 2014), trained on a large source dataset, on to the target dataset such that the re-trained generator imbibes the 'style' of the target while retaining the 'variability' of the source domain. In other words, the re-training is geared towards reducing the infamous problem of the catastrophic forgetting that bogs the realm of transfer learning (Mc-Closkey & Cohen, 1989). While the aforementioned methods show good progress towards adapting a pretrained GAN, there are still shortcomings such as lack of diversity due to over-fitting. Further, these methods require de-novo re-training on every new target, which possibly leads to catastrophic forgetting. In this paper, we intend to tackle some of these issues by addressing the following ques-

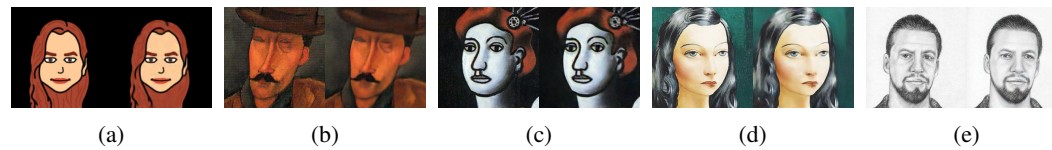

|     |     |     |     |     |
|:---:|:---:|:---:|:---:|:---:|
| (a) | (b) | (c) | (d) | (e) |

Figure 1: (a) Emoji (Hua et al., 2017), (b) Amedeo Modigliani's Art (Yaniv et al., 2019), (c) Fernand Leger's Art (Yaniv et al., 2019), (d) Moise Kisling's Art (Yaniv et al., 2019), (e) Sketches (Wang & Tang, 2009). The left image in each pair is the original image, the right image is the image reconstructed using its embedding in the extended intermediate latent space of StyleGAN2 (Karras et al., 2020b) trained on FFHQ dataset. The latent space accommodates a wide array of data.

tion - **Can a GAN trained on a single large-scale source dataset be adapted to multiple target domains containing very few examples without re-training the pretrained source generator?**

## 1.2 MOTIVATION AND CONTRIBUTIONS

Recently, it has been observed that the latent space of high-fidelity GANs (e.g. StyleGAN2 (Karras et al., 2020b)) is versatile as it can 'accommodate' a large variety of out-of-distribution data (Abdal et al., 2019; 2021; Richardson et al., 2021; Tov et al., 2021). In other words, given a pretrained GAN on a source dataset (source-GAN) and samples from a certain target distribution, the corresponding representations of them can be found out in the latent space of the source-GAN (Abdal et al., 2019; 2021; Richardson et al., 2021; Tov et al., 2021). For instance, Fig. 1 presents images from several target distributions and corresponding reconstructed images using embeddings from the latent space of a StyleGAN2 trained on a large-scale source dataset (FFHQ). It is seen that a wide range of out-of-distribution samples can be embedded in the latent space of source generators. This motivates us to hypothesize the existence of a target-data manifold in the latent space of the source-GAN.

To achieve the aforementioned objective, one straightforward way is to re-train the source-GAN with custom regularization as in (Li et al., 2020). However, these methods are prone to over-fitting when the target data has very few samples (of the order of tens). To alleviate these issues, we propose to find the latent vectors that generate the target data on the fly during the inference without the need to re-train the source-generator. This is accomplished by solving an inference-time optimization problem on the latent space of a pretrained GAN. Recent works (Zhang et al., 2020; Pandey et al., 2021; Wu et al., 2019) have shown the advantage of inference-time latent optimization for several tasks which motivates us to explore the use of inference-time optimization for few-shot generation. We list the contributions of this work below:

1. We propose a simple procedure to utilize a GAN trained on large-scale source-data to generate samples from a target domain with very few (1-10) examples.

2. Our procedure is shown to be capable of generating data from multiple target domains using a single source-GAN without the need for re-training or fine-tuning it.

3. Extensive experimentation shows that our method generates diverse and high-quality target samples with very few examples surpassing the performance of the baseline methods.

## 2 RELATED WORK

**Few shot generative domain adaptation:** In 'generative domain adaptation', a base model pre-trained on source domain is adapted to a related target domain by using few examples. Generally, this is done by re-training the model on the target data via appropriate losses. For example, the authors of Transfer-GAN (Wang et al., 2018) demonstrated that fine-tuning from a single pretrained GAN (Goodfellow et al., 2014) is beneficial for domains with scarce data. Later, the authors in (Noguchi & Harada, 2019) observed that this technique leads to mode collapse, and hence they only fine-tune the scale and shift parameters of the generator. However, this may limit the flexibility of the network. To address this concern, the authors in MineGAN (Wang et al., 2020b) prepend a miner network to the generator to transform the input latent space modeled by multivariate normal distri-bution so that the generated images resemble the target domain. They propose a two step-training

procedure that trains the miner network first, and then the entire pipeline is re-trained using the target data. (Li et al., 2020) adopts Elastic Weight Consolidation (EWC) (Kirkpatrick et al., 2017) to penalize large deviation of the important weights (estimated using Fisher information) while fine-tuning a GAN pretrained on a source domain. (Mo et al., 2020) demonstrates that simple fine-tuning of GANs by freezing the lower layers of the discriminator generates diverse good images in the low data regime. (Zhao et al., 2020a) demonstrates that the filters in the layers close to the observations of both the generator and discriminator of pretrained GANs can be transferred to facilitate generation in a perceptually distinct target domain with limited training data. Further, they propose adaptive filter modulation (AdaFM) to adapt the transferred filters to the target domain. (Ojha et al., 2021) leverage cross-domain correspondence while fine-tuning the source-model. (Lee et al., 2021) uses pairs of positive and negative images from two different domains to learn the cross-domain correspondence. (Xiao et al., 2022) aligns the spatial structural information between the generated image of source and target domain using cross-domain spatial structural consistency loss.

**Few shot augmented generation without pre-training:** In (Karras et al., 2020a), the authors achieve state-of-the-art generation quality with just a few thousand training examples through an adaptive discriminator augmentation mechanism. (Zhao et al., 2020b) generates high-fidelity images using just 100 images utilizing differentiable augmentation (DiffAugment) technique. (Liu et al., 2021) designs a skip-layer channel-wise excitation module and a self-supervised discriminator trained as a feature-encoder to achieve good generation quality using just 100 images. However, the effectiveness of these techniques reduces in extreme few shot e.g., 10-shot settings. In the extreme case, the generative model is trained on a single image (Rott Shaham et al., 2019; Sushko et al., 2021). However, the learned model only manipulates the repeated patterns in that image, fails to generate something that does not exist within the training data.

**Text-guided Domain Adaptation for Image Generators:** StyleGAN-NADA (Gal et al., 2021) develops an interesting approach of CLIP-guided (Radford et al., 2021) training of the generator for zero-shot domain adaptation. (Zhu et al., 2022) builds upon StyleGAN-NADA to develop a one-shot adaptation model and introduces several regularizations to improve generation quality.

**Image-to-Image translation and Style Transfer:** Several methods (Kwong et al., 2021; Pinkney & Adler, 2020; Song et al., 2021) leverage the correspondence between closely related domains and the rich embedding space of StyleGAN2 (Karras et al., 2020b) to perform image-to-image translation. These methods first embed a source domain image into the latent space of StyleGAN2. Next, the latent representation is passed through a model fine-tuned on target data (Song et al., 2021) or a hybrid model obtained by swapping the layers between the source and fine-tuned model (Kwong et al., 2021; Pinkney & Adler, 2020).

Example-based neural style transfer methods (Gatys et al., 2016; Huang & Belongie, 2017; Li et al., 2017) may be employed to transfer the style of a few target domain examples to the plentiful source domain data. However, one single example fails to represent the consistent style across the target domain fully. For instance, the higher-level geometric shape can differ from domain to domain. However, these methods are mostly effective in transferring the color and texture which limits the scope of these methods in generative domain adaptation.

**Uniqueness of our Approach:** Our method is close in spirit to MineGAN (Wang et al., 2020b) that aims to find a latent transformation yielding the target data. However, there are many differences -

1. Proposed method finds point estimates via inference-time optimization compared to (Wang et al., 2020b) where a distribution-level transformation is sought, leading to overfitting with few samples.

2. We train a latent learner every time we desire to generate a batch of images. This contributes to the diversity.

3. This method does not retrain the source generator with target data which preserves the diversity imbided via source-data and enables generation from multiple targets at once.

## 3 PROPOSED METHOD

### 3.1 OVERVIEW

Given a GAN trained on a source dataset and a few samples from the target distribution, our method (Overview in Fig. 2) optimizes a new multi-layer perceptron to output 'novel' target latent vectors. These are then fed to the source-generator to produce novel images from the target domain. Note

that the generator is not a trainable module in our method, it is fixed throughout the training. The only trainable module in our method is of the latent learning network.

## 3.2 BACKGROUND

### 3.2.1 GAN – STYLEGAN

As in previous works Ojha et al. (2021); Xiao et al. (2022), we utilize StyleGAN2 Karras et al. (2020b) for adaptation task. StyleGAN2 consists of two latent spaces: (i) initial latent space, $\mathcal{Z} \subseteq \mathbb{R}^{512}$, and (ii) intermediate latent space, $\mathcal{W} \subseteq \mathbb{R}^{512}$. A 8-layer feed forward network maps $\mathbf{z} \in \mathcal{Z}$ to $\mathbf{w} \in \mathcal{W}$. Manipulating an image needs finding its latent representation. Previous research (Abdal et al., 2019) suggests that a shared latent representation in the $\mathcal{Z}$ or $\mathcal{W}$ space may fail to faithfully embed a given image. Results improve if a separate code is selected for each layer of the StyleGAN. We call this extended intermediate latent space and denote as $\mathcal{W}^+$. We operate on $\mathcal{W}^+$.

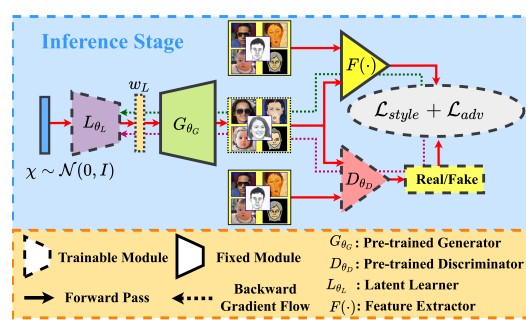

Figure 2: Proposed Model: Given a trained StyleGAN(Karras et al., 2019; 2020b) on a source dataset and a few examples from the target domain, our method trains a latent-learner $L_{\theta_L}$ (a MLP) to generate a new latent code starting from a random Gaussian noise vector. The newly generated latent code is then fed to the source Generator to generate a 'novel' target domain sample. Solid border indicates frozen modules and trainable modules are shown with dotted border. Forward path and backpropagation are shown with solid and dotted lines respectively.

### 3.2.2 PROBLEM FORMULATION

Consider a StyleGAN2 model with generator $G_{\theta_G}$ having parameters $\theta_G$ and discriminator $D_{\theta_D}$ with parameters $\theta_D$. Let $p_S(\boldsymbol{s})$ be the underlying distribution of the source data $\mathcal{S}$ on which it was trained. Let the extended intermediate latent space of $G_{\theta_G}$ be represented by $\mathcal{W}^+$. We are given few samples from target data $\mathcal{T} \sim p_T(\boldsymbol{t})$. Our objective is to use the trained networks $G_{\theta_G}$, $D_{\theta_D}$ and target images $\mathcal{T}$ to generate samples $\mathcal{I} \sim p_T(\boldsymbol{t})$. Note, $G_{\theta_G}(\boldsymbol{w}) \in \mathbb{R}^d$ represents the images generated by the generator $G_{\theta_G}$, and $\boldsymbol{w} \in \mathcal{W}^+ \in \mathbb{R}^m$ as the input from the extended latent space. Note that here $m < d$.

## 3.3 LATENT LEARNING NETWORK

During inference/generation, our objective is to find a latent vector $\boldsymbol{w}_L \in \mathcal{W}^+$ that lies on the target manifold. To achieve this, we use a feed forward network with ReLU activations which we call the latent learner $L_{\theta_L}$ with parameters $\theta_L$. The input to this network is a random vector $\chi$ where $\chi \sim \mathcal{N}(0, I)$ is a sampled from the normal distribution with arbitrary dimensions. Therefore, the objective is to obtain a $\boldsymbol{w}_L = L_{\theta_L}(\chi)$ such that $G_{\theta_G}(\boldsymbol{w}_L) \sim p_T(\boldsymbol{t})$.

To train the latent learner $L_{\theta_L}$, we enforce properties of target domain on $G_{\theta_G}(\boldsymbol{w}_L)$ using appropriate losses involving two components. The first component is the standard style loss (Gatys et al., 2016) between $\boldsymbol{t}$ and $G_{\theta_G}(\boldsymbol{w}_L)$ to capture the target domain features from the image. The second component of the loss function is an adversarial loss: the discriminator $D_{\theta_D}$ of StyleGAN2 acts as a critic which aims to discriminate between generated image $G_{\theta_G}(\boldsymbol{w}_L)$ and original target image $\boldsymbol{t} \in \mathcal{T}$, whereas $L_{\theta_L}$ aims to fool the discriminator, thus making the generated image more like that of the target domain. Note, here we are fine-tuning the discriminator $D_{\theta_D}$ along with the training of $L_{\theta_L}$ while the source generator $G_{\theta_G}$ is untouched (Fig. 2). Our training objective is as follows:

$$\theta_L^*, \theta_D^* = \arg\min_{\theta_L} \max_{\theta_D} \left( \mathcal{L}_{style} + \mathcal{L}_{adv} \right) \tag{1}$$

$$\mathcal{L}_{style} = \mathbb{E}_{\chi, \boldsymbol{t}} \left[ \sum_l \frac{\beta_l}{R_l C_l} ||A^{l;G_{\theta_G}(L_{\theta_L}(\chi))} - A^{l;\boldsymbol{t}}||_2^2 \right] \tag{2}$$

$$\mathcal{L}_{adv} = \mathbb{E}_{\boldsymbol{t}} \left[ \log D_{\theta_D}(\boldsymbol{t}) \right] + \mathbb{E}_{\chi} \left[ \log(1 - D_{\theta_D}(G_{\theta_G}(L_{\theta_L}(\chi)))) \right] \tag{3}$$

Here, $\mathcal{L}_{style}$ is the style loss, $R_l$ and $C_l$ are dimensions of feature map of $l^{th}$-layer, $A^l$ denotes gram matrix of corresponding feature maps as shown in (Gatys et al., 2016). We have used VGG-19 model

($F(.)$ in Fig. 1) to get the feature maps. $\beta_l$ is the hyperparameter corresponding to $l^{th}$-layer, which is set to 1 for all $l$. $\mathcal{L}_{adv}$ is the adversarial loss explained above. While training, generator parameters $\theta_G$ are kept constant, but the gradient is propagated to the latent learner. To update the latent learner, $L_{\theta_L}$, we use non-saturating adversarial loss as outlined in (Goodfellow et al., 2014). Also note that $\mathcal{L}_{style}$ in Equation 1 does not contribute to the maximization step but only to the minimization step of the min-max game. Once the latent learner converges, its output $\boldsymbol{w}_L^* = L_{\theta_L^*}(\mathcal{X})$ is used as the input to the generator to obtain the final generated target images $\mathbf{t}^* = G_{\theta_G}(\boldsymbol{w}_L^*)$.

### 3.3.1 INFERENCE-STAGE LATENT LEARNER RETRAINING

We train the latent learner afresh during inference-stage, whenever we desire to generate a batch of images from the target domain. Note that the same set of target samples are used for training $L_{\theta_L}$ every time, albeit it is trained afresh with different initialization. This would ensure that $L_{\theta_L}$ would not overfit on the few target data points (unlike in (Wang et al., 2020b)), thus inducing diversity in the generated samples. This procedure nevertheless, comes at the cost of an increased inference time which is manageable in many practical settings since the latent learner is a simple 3-layer MLP.

---

**Algorithm 1:** Proposed Method's Pseudo Code

**Input:** Few target domain images $\mathcal{T}$,
   A pretrained generator $G_{\theta_G}$ and
   Discriminator $D_{\theta_D}$
**Output:** A batch of latent vectors $\boldsymbol{w}_L^*$
   and corresponding generated
   images $\mathbf{t}^* = G_{\theta_G}(\boldsymbol{w}_L^*) \in \mathcal{T}$

---

1
2 Sample $\mathcal{X} \sim \mathcal{N}(0, I)$
3 **while** *not converged* **do**
4 $\quad \mathcal{L} \leftarrow \mathcal{L}_{style} + \mathcal{L}_{adv}$
5 $\quad \theta_D^* \leftarrow \theta_D^* - \nu \nabla_{\theta_D^*} \mathcal{L}$
6 $\quad \theta_L^* \leftarrow \theta_L^* - \nu \nabla_{\theta_L^*} \mathcal{L}$
7 Generate $\boldsymbol{w}_L^* = L_{\theta_L^*}(\mathcal{X})$
8 Generate a batch of target images $\mathbf{t}^* = G_{\theta_G}(\boldsymbol{w}_L^*)$

Figure 3: Pseudo code for training.

### 3.4 THEORETICAL DISCUSSIONS

In this section, we discuss possible theoretical justification for the proposed inference time optimization framework. Our hypothesis is mostly based on theory developed in Arjovsky & Bottou (2017). The summary of our discussion is that there exist infinitely many optimal discriminators for the min-max problem in Eq. 1. Hence, due to different sources of stochasticity in the proposed method, the algorithm could latch on to any of the optimal discriminators leading to different latent learner networks, during each instance of inference. To make the argument more formal, let $\boldsymbol{w}_T$ denote the random variable for embedding of the target domain samples $\mathcal{T}$. Also, let $\boldsymbol{w}_L = L_{\theta_L^*}(\mathcal{X})$ denote the random variable for the latent vectors generated by the latent learner $L_{\theta_L^*}$ upon convergence. Further, let $\mathcal{M}_T$ and $\mathcal{M}_L$ denote the manifolds spanned by $G_{\theta_G}(\boldsymbol{w}_T)$ and $G_{\theta_G}(\boldsymbol{w}_L)$ respectively.

Now, as $m < d$ (refer to Section 3.2.2), we have $dim(\boldsymbol{w}_T) < dim(G_{\theta_G}(\boldsymbol{w}_T))$ and $dim(\boldsymbol{w}_L) < dim(G_{\theta_G}(\boldsymbol{w}_L))$, hence $\mathcal{M}_T$ and $\mathcal{M}_L$ will lie in a subspace of dimension less than $d$. In other words, $G_{\theta_G}(\boldsymbol{w}_T)$ and $G_{\theta_G}(\boldsymbol{w}_L)$ do not have full dimension. Also, $\mathcal{M}_T$ and $\mathcal{M}_L$ will not align perfectly almost surely. From this we can conclude that $\mathcal{L} = \mathcal{M}_T \cap \mathcal{M}_L$ has zero measure under both $\mathcal{M}_T$ and $\mathcal{M}_L$ (Lemma 1, Lemma 2 and, Lemma 3 Arjovsky & Bottou (2017)). From Theorem 2.2 of Arjovsky & Bottou (2017), there exists a discriminator, $D_{\theta_D^*} : \mathbb{R}^d \to [0, 1]$ that would optimally discriminate between the samples of the generated and real data distributions. Now we hypothesise that due to the stochasticity originating from different sources (such as initialization, SGD, etc.), the proposed method will discover different manifold $\mathcal{M}_L$ at each instance in inference, leading to a different optimal discriminator $D_{\theta_D^*}$. This in turn leads to a different $L_{\theta_L^*}$ during each inference-time optimization. The aforementioned effect is pronounced especially in the case of extremely low target data regime. This is because every novel inference time optimization routine facilitates the $L_{\theta_L^*}$ network to quickly latch on to a new manifold, $\mathcal{M}_L$. We believe that this serves as a plausible explanation for the diversity in the generated samples that is observed in each inference time of our approach. Developing these ideas further, including formally incorporating the effect of number of target samples on the quality of generation, is a direction for future work.

### 3.5 IMPLEMENTATION DETAILS

The proposed approach is but inference time optimization. During inference, we solve an optimization problem in the parameter space of the latent learner using a combination of style loss (Equation 2) and adversarial loss (Equation 3). We use VGG-19 model to calculate style loss; the layers used to extract feature maps are - {conv1_2, conv2_2, conv3_4, conv4_4, conv5_4}. The latent learner

is a 3-layer MLP with 512 neurons in each layer throughout all the experiments unless otherwise specified. The hidden layers employ 'ReLU' activation, and the final layer has no activation. All the few-shot target samples are used in a single batch (of size 8) for computing the style and adversarial losses. The pseudo-code for the entire algorithm is shown in Algorithm 1 (Fig. 3).

# 4 EXPERIMENTS AND RESULTS

## 4.1 DATASETS

Following previous work (Li et al., 2020; Ojha et al., 2021), we consider Flickr Faces HQ (FFHQ) (Karras et al., 2019) as one of the source domain datasets and adapt to the following target domains — (i) FFHQ-Babies (Ojha et al., 2021), (ii) FFHQ-Sunglasses (Ojha et al., 2021), (iii) face sketches (Wang & Tang, 2009), (iv) emoji faces from `bitmoji.com` API (Taigman et al., 2016; Hua et al., 2017), and (v) portrait paintings from the artistic faces dataset (Yaniv et al., 2019). Next, we consider LSUN Church (Yu et al., 2015) as source domain and adapt to (i) haunted houses (Ojha et al., 2021), and (ii) Van Gogh's house paintings (Ojha et al., 2021). As in previous works (Li et al., 2020; Ojha et al., 2021), we also consider combinations of unrelated source and target domains (FFHQ $\rightarrow$ haunted house and Church $\rightarrow$ face sketches) to probe if the proposed method can adapt successfully when the source and target domains are not 'close' (cf. supp.). We work with $256 \times 256$ images for both source and target domain. All our experiments consider 10 randomly sampled target samples as in previous work unless otherwise specified. For source models, we reuse a pretrained model (Seonghyeon, 2019) for FFHQ at $256 \times 256$ resolution and the official pretrained checkpoint(Karras & Hellsten, 2019) for LSUN Church.

## 4.2 METHODOLOGY, METRICS AND BASELINES

**Methodology:** Even though very few (1, 5, or 10) target examples are used in the method for adaptation, the evaluation is conducted on a larger target set. For example, there are approximately 300, 2500, 2700 and unlimited examples in the sketches (Wang & Tang, 2009), FFHQ-Babies (Ojha et al., 2021), FFHQ-Sunglasses (Ojha et al., 2021), and emoji (Taigman et al., 2016; Hua et al., 2017) datasets, respectively. In such settings, we use the entire target dataset (10000 samples for emoji) for evaluation purposes by generating so many targets using our method. All the mentioned datasets have a dimension of $256 \times 256$, results on higher resolution images can be found in appendix.
**Metrics:** We compute and report the widely used Fréchet Inception Distance (FID) (Heusel et al., 2017) for measuring the similarity of the generated images to the real ones. A lower FID score indicates high similarity. However, being a uni-dimensional score, FID cannot disentangle the two aspects of sample quality and diversity. To alleviate this issue, (Naeem et al., 2020) proposed density and coverage metrics to quantify quality and diversity, respectively. Density is unbounded, and a higher density score indicates better quality. Coverage is bounded by 1, and a higher coverage score is preferred. Another metric that we utilize is the Learned Perceptual Image Patch Similarity (LPIPS) (Zhang et al., 2018) metric, which gives an idea about overfitting on the small amount of target data as shown in (Ojha et al., 2021).
**Baselines:** We compare our method against the following baselines (discussed in Sec. 2) - Transferring GAN (TGAN) (Wang et al., 2018), Batch Statistics Adaptation (BSA) (Noguchi & Harada, 2019), MineGAN (Wang et al., 2020b), Freeze-D (Mo et al., 2020), Non-leaking Adaptive Data Augmentation (Karras et al., 2020a; Zhao et al., 2020b) (TGAN + ADA), Elastic Weight Consolidation (EWC) (Li et al., 2020), Few-shot Image Generation via Cross-domain Correspondence (CDC) (Ojha et al., 2021), $\mathcal{C}^3$: Contrastive Learning for Cross-domain Correspondence (Lee et al., 2021)[1] and Relaxed Spatial Structural Alignment (Xiao et al., 2022).

## 4.3 RESULTS

As can be seen from Table 1, our method outperforms the current state-of-the-art methods on all of the four datasets as measured by FID. From Table 2, it can be seen that the proposed method achieves the best density and coverage scores, indicating the generated samples are diverse and of high quality.

---

[1] We have taken the numbers from the paper itself, as the code is not yet published by the authors. For the same reason, we are unable to present qualitative comparison with this method.

Table 1: FID scores ($\downarrow$) for different target datasets.

| Method | Babies | Sunglasses | Sketches | Bitmoji |
|---|---|---|---|---|
| TGAN (Wang et al., 2018) | $104.79 \pm 0.03$ | $55.61 \pm 0.04$ | $53.41 \pm 0.02$ | $66.69 \pm 0.05$ |
| TGAN + ADA (Karras et al., 2020a) | $102.58 \pm 0.12$ | $53.64 \pm 0.08$ | $66.99 \pm 0.01$ | $68.71 \pm 0.07$ |
| BSA (Noguchi & Harada, 2019) | $140.34 \pm 0.01$ | $76.12 \pm 0.01$ | $69.32 \pm 0.02$ | $105.56 \pm 0.03$ |
| FreezeD (Mo et al., 2020) | $110.92 \pm 0.02$ | $51.29 \pm 0.05$ | $46.54 \pm 0.01$ | $71.16 \pm 0.01$ |
| MineGAN (Wang et al., 2020b) | $98.23 \pm 0.03$ | $68.91 \pm 0.03$ | $64.34 \pm 0.02$ | $86.40 \pm 0.04$ |
| EWC (Li et al., 2020) | $87.41 \pm 0.02$ | $59.73 \pm 0.04$ | $71.25 \pm 0.01$ | $73.82 \pm 0.04$ |
| CDC (Ojha et al., 2021) | $74.39 \pm 0.03$ | $42.13 \pm 0.04$ | $45.67 \pm 0.02$ | $69.54 \pm 0.05$ |
| $\mathcal{C}^3$ (Lee et al., 2021) | $67.55 \pm 2.23$ | $36.69 \pm 2.63$ | $41.50 \pm 1.64$ | — |
| RSSA (Xiao et al., 2022) | $75.67 \pm 0.39$ | $44.35 \pm 0.06$ | $54.58 \pm 0.51$ | $67.14 \pm 0.63$ |
| Proposed Method | $\mathbf{63.31 \pm 0.05}$ | $\mathbf{35.64 \pm 0.15}$ | $\mathbf{35.59 \pm 0.13}$ | $\mathbf{64.50 \pm 0.12}$ |

Table 2: Comparison of Density ($\uparrow$) and Coverage ($\uparrow$) scores for FFHQ babies and sketches datasets.

| Method | Babies | | Sketches | |
|---|---|---|---|---|
| | Density | Coverage | Density | Coverage |
| TGAN (Wang et al., 2018) | 0.379 | 0.250 | 0.221 | 0.401 |
| TGAN + ADA (Karras et al., 2020a) | 0.434 | 0.285 | 0.193 | 0.374 |
| FreezeD (Mo et al., 2020) | 0.418 | 0.217 | 0.415 | 0.436 |
| MineGAN (Wang et al., 2020b) | 0.803 | 0.125 | 0.394 | 0.263 |
| EWC (Li et al., 2020) | 0.301 | 0.325 | — | — |
| CDC (Ojha et al., 2021) | 0.690 | 0.467 | 0.149 | 0.492 |
| RSSA (Xiao et al., 2022) | 0.961 | 0.402 | 0.070 | 0.691 |
| Proposed Method | **1.118** | **0.611** | **0.445** | **0.716** |

Table 3: Intra-cluster pairwise LPIPS distance ($\uparrow$)

| Method | Amedeo's Paintings | Sketches |
|---|---|---|
| TGAN (Wang et al., 2018) | $0.41\pm0.03$ | $0.39\pm0.03$ |
| TGAN + ADA (Karras et al., 2020a) | $0.51\pm0.04$ | $0.41\pm0.05$ |
| BSA (Noguchi & Harada, 2019) | $0.39\pm0.04$ | $0.35\pm0.01$ |
| FreezeD (Mo et al., 2020) | $0.40\pm0.03$ | $0.39\pm0.03$ |
| MineGAN (Wang et al., 2020b) | $0.42\pm0.03$ | $0.40\pm0.05$ |
| EWC (Li et al., 2020) | $0.52\pm0.03$ | $0.42\pm0.03$ |
| CDC (Ojha et al., 2021) | $0.60\pm0.01$ | $0.45\pm0.02$ |
| $\mathcal{C}^3$ (Lee et al., 2021) | — | $0.45\pm0.03$ |
| RSSA (Xiao et al., 2022) | $0.53\pm0.07$ | $0.43\pm0.05$ |
| Proposed Method | $\mathbf{0.61\pm0.02}$ | $\mathbf{0.48\pm0.02}$ |

As in (Ojha et al., 2021), we assign 1000 generated images to one of the $k$ possible clusters (for $k$-shot generative domain adaptation, $k = 10$ in Table 3) based on the lowest LPIPS distance (Zhang et al., 2018). Next, we compute the average pair-wise LPIPS metric among the members of the same cluster. Finally, we take the average over the $k$ clusters. A method will have a zero score if it reproduces the original images.

A lower value of this metric implies the generated images are similar or, in other words, less diverse. As can be seen from Table 3, our proposed method achieves the best intra-cluster pair-wise LPIPS distance for Amedeo Modigliani's paintings and sketches datasets. Figure 4 and 5 depicts a few examples of generated targets from our method and the previous SoTA (Ojha et al., 2021; Xiao et al., 2022). It can be seen that our method can generate diverse examples on multiple target domains. (See appendix for more qualitative results on different target domains. Appendix also contains examples of editing using the learnt representations.)

Table 4: Ablation studies to understand the impact of different loss terms.

| Loss Components | | FID Score ($\downarrow$) | |
|---|---|---|---|
| Adv. Loss | Sty. Loss | Babies | Sketches |
| ✓ | ✗ | 69.78 | 68.99 |
| ✗ | ✓ | 212.87 | 139.12 |
| ✓ | ✓ | **62.14** | **35.59** |

Table 5: Impact of the capacity of the $L_{\theta_L}$ on generation quality.

| Latent learner Capacity | FID Score ($\downarrow$) | |
|---|---|---|
| | Babies | Sketches |
| Small Network Trainable parameters: 787,456 | 66.28 | **35.22** |
| Medium Network Trainable parameters: 1,050,112 | **62.14** | 35.59 |
| Large Network Trainable parameters: 2,624,000 | 64.43 | 38.61 |

## 4.4 ABLATION STUDIES

**Loss Components:** Table 4 presents the impact of the two loss components using two datasets FFHQ-Babies and Sketches. The best FID score is achieved when both the loss components are present. Further, the style loss acts more as a regularizer, whereas adversarial loss improves quality.

Table 6: Effect of $k$ in $k$-shot adaptation on generation quality as measured by FID score ($\downarrow$).

| Method | 1-shot | | 5-shot | | 10-shot | |
|---|---|---|---|---|---|---|
| | Babies | Sketches | Babies | Sketches | Babies | Sketches |
| CDC (Ojha et al., 2021) | 105.58 | 81.95 | 73.63 | 51.01 | 74.39 | 45.67 |
| RSSA (Xiao et al., 2022) | 157.84 | 119.66 | 96.42 | 63.34 | 75.67 | 54.58 |
| Proposed Method | **105.13** | **79.20** | **65.47** | **41.88** | **62.14** | **35.59** |

$L_{\theta_L}$ **Capacity:** Next, to understand the impact of the complexity of the latent learner (overfitting vs underfitting) on generation quality, in Table 5, we consider three capacities for the latent learner. The smaller capacity $L_{\theta_L}$ is a 2-layer MLP with 512 neurons in each layer, the larger capacity $L_{\theta_L}$ is again a 3-layer MLP with 1024 neurons in the hidden layers and 512 neurons in the output layer, and

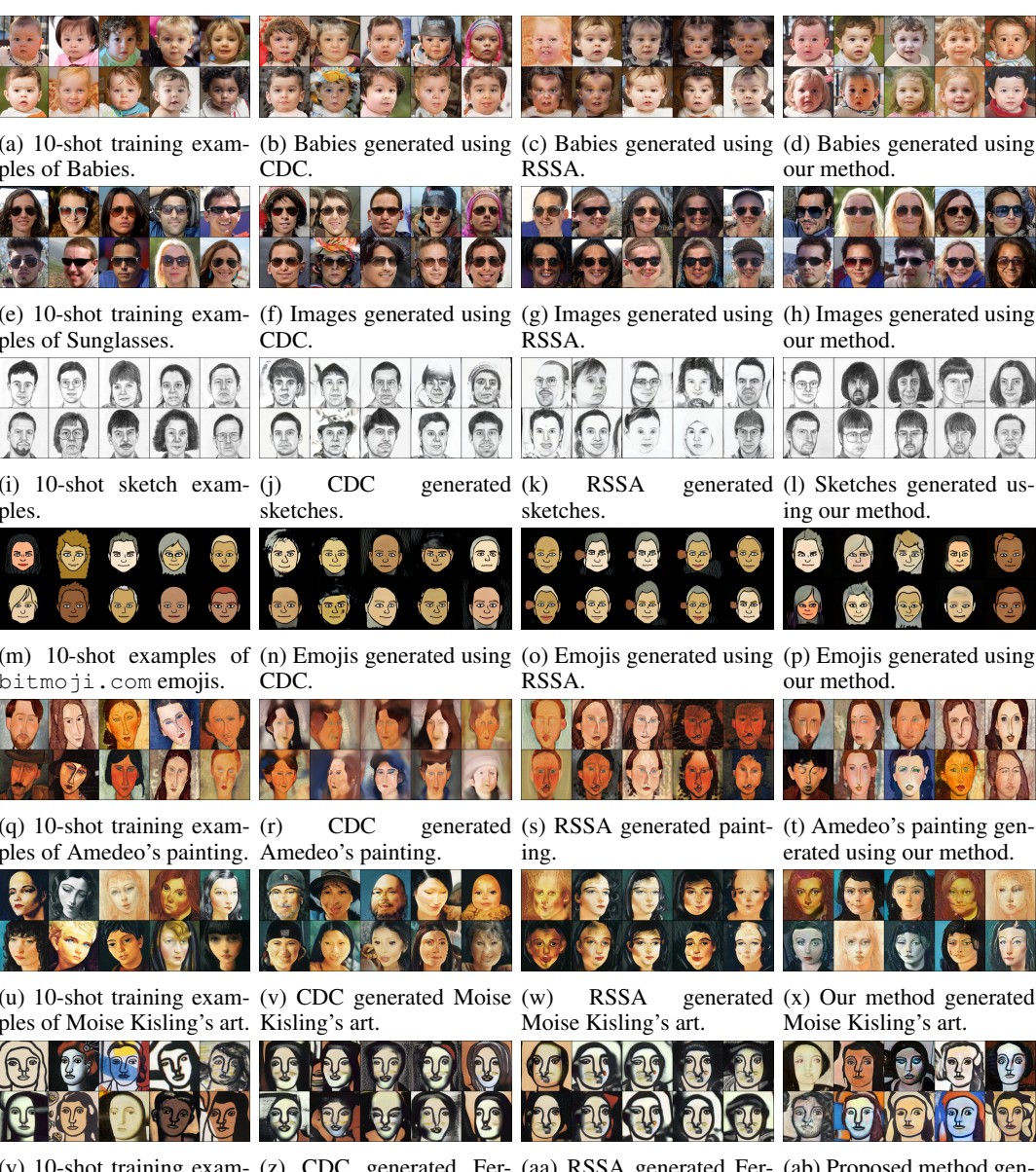

(a) 10-shot training examples of Babies.

(b) Babies generated using CDC.

(c) Babies generated using RSSA.

(d) Babies generated using our method.

(e) 10-shot training examples of Sunglasses.

(f) Images generated using CDC.

(g) Images generated using RSSA.

(h) Images generated using our method.

(i) 10-shot sketch examples.

(j) CDC generated sketches.

(k) RSSA generated sketches.

(l) Sketches generated using our method.

(m) 10-shot examples of bitmoji.com emojis.

(n) Emojis generated using CDC.

(o) Emojis generated using RSSA.

(p) Emojis generated using our method.

(q) 10-shot training examples of Amedeo's painting.

(r) CDC generated Amedeo's painting.

(s) RSSA generated painting.

(t) Amedeo's painting generated using our method.

(u) 10-shot training examples of Moise Kisling's art.

(v) CDC generated Moise Kisling's art.

(w) RSSA generated Moise Kisling's art.

(x) Our method generated Moise Kisling's art.

(y) 10-shot training examples of Fernand Leger's art.

(z) CDC generated Fernand Leger's art.

(aa) RSSA generated Fernand Leger's art.

(ab) Proposed method generated Fernand Leger's art.

Figure 4: Source: FFHQ. Each row represents one target domain among - Babies, Sunglasses, Sketches, Emoji, Amedeo Modigliani's paintings, Moise Kisling's art, and Fernand Leger's art. The $1^{st}$ column presents 10-shot target examples. The $2^{nd}$, $3^{rd}$ and $4^{th}$ columns present 10 images generated by CDC (Ojha et al., 2021), RSSA (Xiao et al., 2022), and proposed method respectively.

medium capacity is a 3-layer MLP with 512 neurons in each layer. It is seen, the medium-capacity network used across all the experiments achieves the optimum performance.

**Number of Target Examples:** Till now, the results presented utilize 10 target domain images for generative domain adaptation. Table 6 presents the impact of lesser training data on generation quality. We compare the proposed method with the current SOTA methods (Ojha et al., 2021; Xiao et al., 2022). It is seen, the generation quality improves with $k$, and the proposed method performs considerably better in almost all cases considered. Refer to the appendix for qualitative analysis.

## 4.5 APPLICATION: IMAGE TRANSLATION

One potential application of our method is to transfer the style of the image of one domain to another domain. Figure 6 presents some examples where source domain (FFHQ) images have been trans-

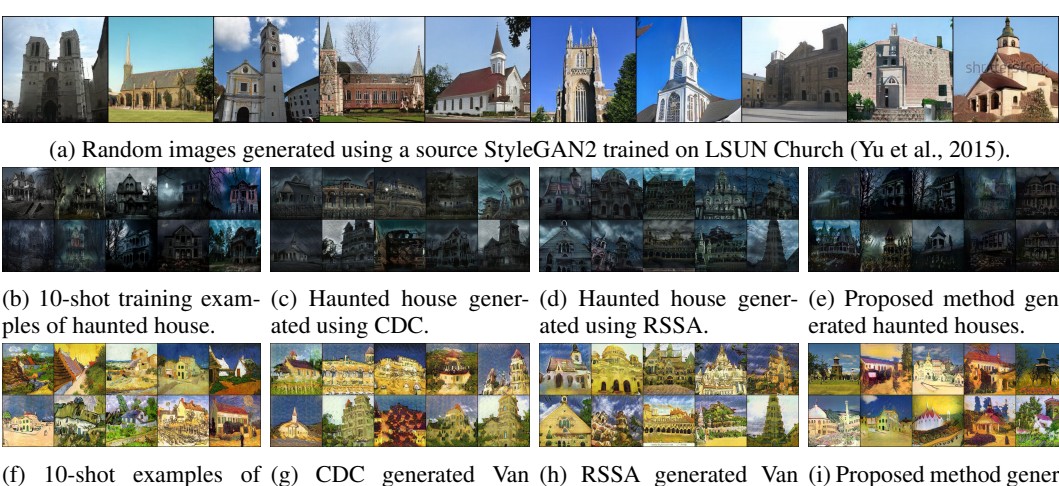

(a) Random images generated using a source StyleGAN2 trained on LSUN Church (Yu et al., 2015).

(b) 10-shot training exam- (c) Haunted house gener- (d) Haunted house gener- (e) Proposed method gen-
ples of haunted house. ated using CDC. ated using RSSA. erated haunted houses.

(f) 10-shot examples of (g) CDC generated Van (h) RSSA generated Van (i) Proposed method gener-
Van Gogh's paintings. Gogh style paintings. Gogh style paintings. ated Van Gogh paintings.

Figure 5: Source: LSUN Church (Yu et al., 2015). Target: Haunted house (Ojha et al., 2021) and Van Gogh's house paintings (Ojha et al., 2021). The $1^{st}$ column of $2^{nd}$ and $3^{rd}$ row presents 10-shot examples from the target domains. The $2^{nd}$, $3^{rd}$, and $4^{th}$ columns present 10 images generated using CDC (Ojha et al., 2021), RSSA (Xiao et al., 2022), and proposed method respectively.

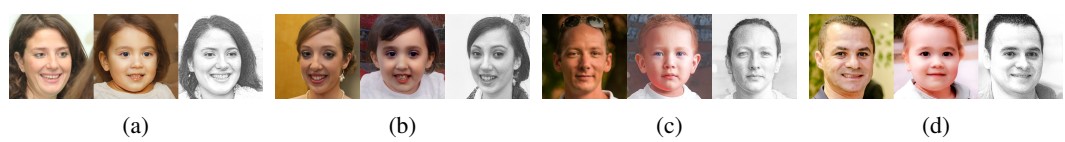

(a)            (b)            (c)            (d)

Figure 6: Transferring source domain (FFHQ) image (left most image in each sub-figure) to target domains Babies (center image) and Sketches (right most image).

lated to two target domains — babies and sketches. To do this, we use the embedding of the source image and target images in place of random input noise to the latent learner. Further, we incorporate two additional loss terms: (i) Multi-scale SSIM loss between the source image and generated image, (ii) Structural loss between source, target, and generated images. Please note, as before, we keep the source generator fixed and update only the latent learner parameters. See appendix for more details.

## 5    LIMITATIONS, RISK AND CONCLUSION

We conclude this article with a discussion on limitation of this work and possible future directions.
**Limitation:** In our method, inference time optimization mandates that we solve an optimization problem every time we want to generate new images, which may increase the inference time for certain applications. One way to address this issue is to store the outputs of the latent learner as we solve inference-time optimization and train another generative model to sample from the manifold of interest in the latent space. To validate the proof of this concept, we trained a WGAN (Arjovsky et al., 2017; Gulrajani et al., 2017) to generate latent codes for the babies and sunglasses datasets and achieved comparable FID scores. We intend to probe this direction further in our future work.
**Risk:** One of the possible negative social impacts of deep generative frameworks is the creation of deepfakes (Vaccari & Chadwick, 2020). However, significant amount of work (Wang et al., 2020a; Dolhansky et al., 2020; Mirsky & Lee, 2021) have been done to tackle the problem of identifying real vs. fake images to counter deepfakes.
**Conclusion:** In this work, we present a methodology to build data-efficient generative models, demonstrating that existing source models can be used as it is (without retraining or fine-tuning) to model new distributions with less data. We believe that our work is an important step towards few-shot 'generative domain adaptation' where we have demonstrated that the same source generator can be utilized effectively to generate source domain samples and multiple target domain samples. Additionally, our proposed method can be viewed as a step towards continual learning for generative task where the same generator can generate data from different domains.

## REPRODUCIBILITY STATEMENT

To ensure that the proposed work is reproducible, we have included an Algorithm (Refer to Fig. 3). We have clearly mentioned different loss terms (Refer to Eq. 1, 2 and 3). Further, we have an explicit section (Refer to Sec. 3.5) on implementation details. The code of the proposed method is available at: `https://github.com/arnabkmondal/GenDA`. This is an anonymous link which doesn't reveal author identity. For convinience, we have also included a ReadMe file using which our results can be reproduced.

## ACKNOWLEDGMENTS

We thank IIT Delhi HPC facility[2] for computational resources. Parag Singla is supported by IBM AI Horizon Networks (AIHN) grant and IBM SUR awards. Any opinions, findings, conclusions or recommendations expressed in this paper are those of the authors and do not necessarily reflect the views or official policies, either expressed or implied, of the funding agencies.

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
