# OpenReview forum: "Few-shot Cross-domain Image Generation via Inference-time Latent-code Learning"
_ICLR.cc/2023/Conference — ICLR 2023 notable top 25%_

### Official Review · Reviewer_G9qY · 2022-10-24

**Confidence:** 2
**Correctness:** 3
**Technical Novelty And Significance:** 3
**Empirical Novelty And Significance:** 3
**Recommendation:** 8

**Clarity, Quality, Novelty And Reproducibility:**

- The paper has good writing quality and clarity.
- Although the proposed method looks simple and incremental to previous studies, the numerical results have shown the effectiveness and efficiency of the proposed method.
- The code is anonymously provided for better reproducibility.

**Strength And Weaknesses:**

*Strengths*
- The presentation quality of the paper is of good quality: the proposed method is well stated.
- Comparison with previous literature is well studied.
- The proposed method is novel in that it provides an intuitive yet efficient way to reduce overfitting and increase diversity.
- Numerical studies and comparisons with previous methods have been well conducted. The results show that the proposed method is at least comparable to or better than state-of-the-art methods.
*Weaknesses*
- Although this paper is experiment-biased, it could be more plausible to include theoretical discussions on why the introduction of the latent space learner could help reduce overfitting and increase diversity.

**Summary Of The Paper:**

This paper proposes a novel method to generate realistic images on the target sparse domain by transferring a large-scale generator from the source domain without retraining the generator. Also, the proposed method can generate diverse target samples. The key idea is to solve an optimization problem on the latent domain of the generator at inference time. The proposed methodology is clearly stated and extensive comparisons with state-of-the-art methods have been studied. The proposed method has shown very competitive numbers in the experiments.

**Summary Of The Review:**

The proposed method shows great potential in mitigating the overfitting and diversity issues in solving domain transfer problems. The experiments have shown promising results with the proposed method. The paper is overall well presented. I would recommend acceptance of the paper.

---

> ### Author Response · Authors · 2022-11-12
> **Reply to Reviewer G9qY's Comments**
>
> We appreciate you taking the time to evaluate and commend our work. To highlight the overview of the edits we have made, we have posted a common comment. We address the particular criticisms and issues you expressed regarding our work in this comment.
>
> *[On theoretical Insights behind our Approach]:*
>
> Based on reviewer’s suggestions, we have added a new Section #3.4 describing the theoretical underpinnings of our approach.
>
> The summary of our discussion is that there exist infinitely many optimal discriminators for the min-max problem in Eq. 1 of the updated manuscript. Hence, due to different sources of stochasticity in the proposed method, the algorithm could latch on to any of the optimal discriminators leading to different latent learner networks, during each instance of inference. The resulting difference in the parameterization of the latent learner during each inference helps preserve the diversity of the generated samples, especially in a very low data setting. For a more formal argument please refer to section 3.4 in the main manuscript.

---

### Official Review · Reviewer_8gcS · 2022-10-24

**Confidence:** 5
**Correctness:** 4
**Technical Novelty And Significance:** 4
**Empirical Novelty And Significance:** 4
**Recommendation:** 8

**Clarity, Quality, Novelty And Reproducibility:**

The text has been written succinctly and the experiments back the text in terms of clear rationale. The quality of the text is quite high and the process is fairly novel. The techniques have been well described and can easily be reproduced in a reasonable amount of GPU resources.

**Details Of Ethics Concerns:**

No Concerns about ethics of the submission.

**Strength And Weaknesses:**

Strengths:

* Simple technique for learning a particular style using a set of few images (~10)
* Easy to train and the training converges in a reasonable time
* Provides an alternative to StyleGAN NADA where the style cannot be explicitly described in a few words

Weaknesses:

* The style loss using Gram Matrix only allows particular kinds of texture based styles to be enabled by this technique
* Geometric transformations such as adding a smile or wink would be harder with this kind of technique.

**Summary Of The Paper:**

The paper provides a simple approach to learn a particular style of generative model using a base model as StyleGAN. It learns a small latent encoder embedding network which encodes the normal distribution into the target latent space. It learns using the style loss proposed by Gatys et al and the StyleGAN Discriminator's adversarial loss. The results are comprehensive and the technique is beneficial for face editing applications.

**Summary Of The Review:**

The simplicity of the approach by learning an encoder to transform normal distribution to the appropriate target latent space of the StyleGAN network provides an interesting approach to face stylization. The work provides the right references as well as compares against well known techniques such as StyleGAN NADA on well backed metrics.

The paper deserves to be accepted to ICLR as it adheres to its high standards.

---

> ### Author Response · Authors · 2022-11-12
> **Reply to Reviewer 8gcS's Comments**
>
> We thank you for your review and appreciation of our work. We have written a common comment to highlight the summary of the revisions that we have made. In this comment we respond to the specific comments/concerns you have raised about our work.
>
> *[Capturing non-Texture based Transformations]:*
>
> We believe that generating images having geometric transformations such as smiles are also possible using our method given a few samples with such transformations. For instance, we have shown results on sunglasses which are akin to images with a wink. We ascribe this to the presence of the adversarial loss in our method in addition to the Gram matrix based style loss.
>
> Further, we did attempt image editing with geometric transformations in figure 6 and 7 of the supplementary material using an unsupervised algorithm SeFa [Shen & Zhou (2021)]. As can be seen in each column of Figure 6a, the neutral face of the generated image (in the middle row) becomes smiling in the top row and serious/fearful in the last row.

---

### Official Review · Reviewer_ubZS · 2022-10-26

**Confidence:** 4
**Correctness:** 4
**Technical Novelty And Significance:** 3
**Empirical Novelty And Significance:** 3
**Recommendation:** 8

**Clarity, Quality, Novelty And Reproducibility:**

This paper is clear and of good quality. I find it to be easy to read and, as far as I'm aware, it is a novel method.

**Strength And Weaknesses:**

Strengths:

- concise and straightforward method for performing the task at hand
- generally well-written and easy to follow
- compelling results for this very-few-shot domain adaptation generation

Weaknesses:

- The paper feels like it could be a bit shortened. The related work and background are longer than I'd expect, and there are fewer qualitative examples than I'd like in the main text.
- I am concerned about this method's generalization to higher number of shots. I know other few-shot works use 30 or 100 examples. (I don't mind needing to retrain the networks for each new generation batch). How does this method perform at different shot capacity?
- I am curious about the effects of latent learner's capacity. What happens with an extremely small model (1 fully connected layer)? When the model is the identity function, do the learned point simply become training set?



**Summary Of The Paper:**

This paper introduces a method to generate samples from a target domain containing very few shots using a fixed pretrained GAN.

**Summary Of The Review:**

The proposed method is simple and intuitive, with strong quantitative/qualitative results. However, the focus on exclusively extremely few shot setting is somewhat limiting, so I advocate for a weak accept.

Post rebuttal: My concerns have been addressed, therefore I increase my score to accept.

---

> ### Author Response · Authors · 2022-11-12
> **Reply to Reviewer ubZS's Comments**
>
> We thank you for your insightful comments about our work. We have added a common comment to highlight the changes we have made in the revised manuscript and the supplementary material as per the feedback received from different reviewers. Here we provide a point-by-point response to the issues raised by you.
>
> *[Shortening of the Paper and Qualitative Examples]:*
>
> We have edited the paper to reduce the background section and added more qualitative examples in Figure 4 (Figure 4u - 4ab). Due to space crunch, additional visual examples on different resolution images and complex datasets are added in the supplement.
>
>
> *[Generalization to Higher number of Shots]:*
>
> We have performed experiments on higher shots (#shots - 100) as suggested by you using the babies dataset. We have presented the results in Table 1 of the supplementary material. We present the findings here as well for easy reference:
>
> | **Method**      | **20-shot** | **100-shot** |
> |-----------------|-------------|--------------|
> | CDC             | 57.69       | 39.32        |
> | RSSA            | 59.14       | 41.56        |
> | Proposed Method | **54.98**      | **39.27**        |
>
> We note that though the relative gain of our method compared to baselines decreases as the number of shots is increased (which is expected, since other approaches can now better fine-tune the source generator), we still outperform the baselines. This points to the robustness of our approach across varying # of shots. We note that the best relative advantage of our approach is seen at a very small number of shots.
>
>
> *[Effect of Latent Learner’s Capacity]:*
>
> — Case A (Extremely small model with 1 layer):
> The only layer is the output layer therefore it does not have any activation, making it a linear transformation. Consequently, the model fails to learn any useful transformation. We have performed this experiment and the generated images do not look anything like the target domain. If you mean a network with one hidden layer, results are already present in Table 5 of the main paper (small network in Table 5 refers to a network with 1 hidden layer, as described in Section 4.4).
>
> — Case B (Model is identity function):
> The model is an identity function meaning the noise is fed directly to the pre-trained generator. Since, there is no appropriate transformation corresponding to the target domain, the generated data will not look like the target domain training set.

---

> > ### Comment · Reviewer_ubZS · 2022-11-18
> > **Thank you, authors**
> >
> > My concerns have been addressed, therefore I increase my score to accept.

---

### Official Review · Reviewer_jAjp · 2022-11-02

**Confidence:** 4
**Correctness:** 4
**Technical Novelty And Significance:** 2
**Empirical Novelty And Significance:** 3
**Recommendation:** 8

**Clarity, Quality, Novelty And Reproducibility:**

This paper has good clarity and quality.

The novelty is not high but quite reproducible and applicable.

**Strength And Weaknesses:**

Strength:
- Highly achievable and reasonable.
  Generally the pain part of GAN is that latent space is always not fully exploitable through training. Existing methods focus too much on
model itself but not build on transfer learning.

Weakness:
- This method still needs some high resolution and complex dataset to demonstrate.

**Summary Of The Paper:**

This paper shows a few shot GAN by embed latent factor into an existing GAN to transfer to a new style. This paper tries to mix pre-trained
GAN with few shot learning.

Their contribution can be concluded as:
-propose a simple procedure to utilize a GAN trained on large-scale source-data to
generate samples from a target domain with very few (1-10) examples.
- Their procedure is shown to be capable of generating data from multiple target domains
using a single source-GAN without the need for re-training or fine-tuning it.
- Extensive experimentation shows that our method generates diverse and high-quality target
samples with very few examples surpassing the performance of the baseline methods.

**Summary Of The Review:**

In this work, authors present a methodology to build data-efficient generative models, demonstrating that existing source models can be used as it is (without retraining or fine-tuning) to model new distributions with less data. The method is not very complicate but it shows a
good direction and very reproducible work as a direction.

---

> ### Author Response · Authors · 2022-11-12
> **Reply to Reviewer jAjp's Comments**
>
> We thank you for reviewing and appreciating our work. We have added a common comment highlighting different changes that we have made based on the feedback provided by all the reviewers. Below, we specifically address the concerns raised by you in the review:
>
> *[Experiments High Resolution and Complex Datasets]:*
>
> To validate whether the method can generalize across different resolutions of images, we have performed several experiments on additional complex datasets.
>
> **Example 1 (1024 x 1024 images) -** We have used the StyleGAN2 trained on the map-dataset (https://github.com/justinpinkney/awesome-pretrained-stylegan2#maps) as the source and floor-plan as the target. In this experiment the images are of size 1024 x 1024. Refer to Figure 10 in the updated supplementary material.
>
> **Example 2 (512 x 512 images) -** We have adapted a source GAN trained to generate cars (https://nvlabs-fi-cdn.nvidia.com/stylegan2/networks/stylegan2-car-config-f.pkl) to target domain wrecked cars (Ojha et al., 2021). Images in this experimental setting are of 512 x 512 resolution. Refer to Figure 9 in the updated supplementary material.
>
> **Example 3 (64 x 64 images) -** To validate our method in lower resolution images, we have performed experiments to adapt 64x64 FFHQ source model to CeleB-A Female dataset using FFHQ source model. We have used 64 x 64 sized images for this experiment. The results can be found in Figure 8 of the updated supplementary material.
>
> The results (Figures 8, 9, and 10 in the updated supplementary material) demonstrate that our approach is effective for transfer across multiple image resolutions, as well as, for complex (source as well target) datasets.

---

### Author Response · Authors · 2022-11-12
**Overview of the Revision**

We thank all the reviewers for thinking positively about our work. We highly appreciate their comments and have tried to the best of our abilities to address all the comments that have helped to further improve our paper. Here is a summary of changes that we have made:

1. Added theoretical discussions (Sec. 3.4) on why the method should work in the updated manuscript (Ref. comments by Reviewer - G9qY)

2. Added new experiments of high as well as low resolution images and additional complex datasets in the supplementary material (Figure 8, 9, and 10). (Ref. comments by Reviewer jAjp)

3. Edited the paper to reduce superfluous text in the background (section 3.2 of the main paper) and added more visual examples in Figure 4 of the main text. Also added results to understand the impact of higher number of shots in the appendix. (Ref. comments by Reviewer ubZS)

The edited text has been highlighted in blue to reduce reviewers’ effort in finding out the changes. The color will be reverted in the final draft.

---

### Decision · Program_Chairs · 2023-01-20

**Decision:**

Accept: notable-top-25%

**Justification For Why Not Higher Score:**

The focus on exclusively extremely few shot setting is somewhat limiting, and I agree.

At most a spotlight paper, could be a poster as well.

**Justification For Why Not Lower Score:**

A clean and simple method which yields a good performance boost.  Theoretical discussion is also supplemented for the interest of the ICLR audiences.

**Metareview: Summary, Strengths And Weaknesses:**

This paper presents a method to generate samples from a target domain containing very few shots using a fixed pretrained GAN.


+ concise and straightforward method for performing the task at hand
+ compelling results for this very-few-shot domain adaptation generation
+ Provides an alternative to StyleGAN NADA where the style cannot be explicitly described in a few words




**Note From Pc:**

if the above contains the word "oral" or "spotlight" please see: "oral" presentation means -> notable-top-5% and "spotlight" means -> notable-top-25%. As stated in our emails, we are disassociating presentation type from AC recommendations

**Summary Of Ac-Reviewer Meeting:**

N/A